# Can Social Innovation Make a Change in European and Mediterranean Marginalized Areas? Social Innovation Impact Assessment in Agriculture, Fisheries, Forestry, and Rural Development

Elisa Ravazzoli [1,*], Cristina Dalla Torre [1,2], Riccardo Da Re [2], Valentino Marini Govigli [3], Laura Secco [2], Elena Górriz-Mifsud [3,4], Elena Pisani [2], Carla Barlagne [5], Antonio Baselice [6], Mohammed Bengoumi [7], Marijke Dijskhoorn-Dekker [8], Arbia Labidi [7], Antonio Lopolito [9], Mariana Melnykovych [3,10], Manfred Perlik [11], Nico Polman [8], Simo Sarkki [12], Achilleas Vassilopoulos [13], Phoebe Koundouri [14], David Miller [5], Thomas Streifeneder [1] and Maria Nijnik [5]

1. Institute for Regional Development Eurac Research, Viale Druso, 1, 39100 Bolzano, Italy; cristina.dallatorre@eurac.edu (C.D.T.); thomas.streifeneder@eurac.edu (T.S.)
2. Department of Land, Environment, Agriculture and Forestry (TESAF), University of Padova, Via dell'Università, 16, 35020 Legnaro, Italy; riccardo.dare@unipd.it (R.D.R.); laura.secco@unipd.it (L.S.); elena.pisani@unipd.it (E.P.)
3. European Forest Institute, Mediterranean Facility (EFIMED), St. Antoni Maria Claret, 167, 08025 Barcelona, Spain; valentino.govigli@efi.int (V.M.G.); elena.gorriz@ctfc.es (E.G.-M.); mariana.melnykovych@wsl.ch (M.M.)
4. Forest Science and Technology Center of Catalonia (CTFC), 25280 Solsona, Spain
5. The James Hutton Institute, Craigiebuckler, Aberdeen AB15 8QH, Scotland, UK; carla.barlagne@hutton.ac.uk (C.B.); david.miller@hutton.ac.uk (D.M.); maria.nijnik@hutton.ac.uk (M.N.)
6. Department of Sciences of Agriculture, Food Natural resources and Engineering, University of Foggia, 71121 Foggia, Italy; antonio.baselice@unifg.it
7. Food and Agriculture Organisation (FAO), Subregional Office for North Africa, les Berges du Lac 1, Tunis 1053, Tunisia; mohammed.bengoumi@fao.org (M.B.); arbia.labidi@fao.org (A.L.)
8. Wageningen Economic Research, Prinses Beatrixlaan 582-528, 2595 BM The Hague, The Netherlands; marijke.dijkshoorn@wur.nl (M.D.-D.); nico.polman@wur.nl (N.P.)
9. Department of Economics, Management and Territory, University of Foggia, 71121 Foggia, Italy; antonio.lopolito@unifg.it
10. Swiss Federal Institute for Forest, Snow and Landscape Research (WSL), 8903 Birmensdorf, Switzerland
11. Centre for Development and Environment, University of Bern, Mittelstrasse 43, 3012 Bern, Switzerland; manfred.perlik@unibe.ch
12. Cultural Anthropology Programme, University of Oulu, 90570 Oulu, Finland; Simo.sarkki@oulu.fi
13. Department of Economics, University of Ioannina, University Campus, 45110 Ioannina, Greece; avas@uoi.gr
14. School of Economics, Athens University of Economics and Business, 76 Patission Street, 104 34 Athens, Greece; pkoundouri@aueb.gr
* Correspondence: elisa.ravazzoli@eurac.edu

**Abstract:** Social innovation (SI) impacts are long-term changes that affect different dimensions of territorial capital (i.e., economy, society, environment, governance) for the territory in which SI occurs. Yet, systematic empirical evidence and theoretically sound assessments of the impacts of SI are scarce. This paper aims to fill the gap and assess the different aspects of SI's impacts in European and Mediterranean areas that are characterized by marginalization processes. To assess the impacts of SI in marginalized areas, we use the evaluation framework developed within the Social Innovation in Marginalized Rural Areas (SIMRA) Horizon 2020 project and apply it to nine SI initiatives related to the fields of agriculture, fisheries, forestry, and rural development. Our findings show that SI produces cross-sectoral (societal, economic, environmental, and governmental) and multi-level impacts (on individuals, community, and society), which have improved the societal well-being, and contributed to the reduction of certain forms of marginality, mainly inside the territory in which SI occurred.

**Keywords:** social innovation; socio-economic impacts; institutional impacts; environmental impacts; societal well-being; European societal challenges; marginalization; sustainability challenges; local level

## 1. Introduction

The concept of social innovation (SI) is an emerging one, both in academia and in practice fields. It refers to initiatives aiming to deliver impacts for enhancing societal well-being. Despite the lack of a commonly accepted definition, there is a general agreement that SI refers to both a process of the transformation of social practices (i.e., attitudes, behaviors, networks of collaboration) and to the outcomes in terms of new products and services (i.e., novel ideas, models, services, and new organizational forms). Both processes and outcomes seek to address complex challenges, and to respond to needs that the market or the public sector are not able to address adequately [1,2] or are seeking alternatives to existing institutions [3–5]. In this paper we aim to show that SI may play an important role in the development of communities characterized by forms of marginalities, such as physical constraints, limited access to roads or infrastructure, and socio-economic factors such as brain drain and social exclusion [6,7]. Here, SI can offer new solutions to complex and urgent problems, promoting sustainable solutions and addressing sustainability challenges and ambitions, such as the transformation of existing structures towards a low carbon society, climate change mitigation, fair distribution of income, sustainable livelihoods, and lifestyles [8–11]. SI's initiatives show their commitments towards the creation of social and environmental value, and the delivery of social benefits to local communities [12]. The concept of SI is also applied to companies engaging in corporate social and environmental practices (e.g., corporate social responsibility, socially responsible investing, corporate social innovation, and green and sustainable finance) [13,14]. Socially and environmentally responsible businesses and SIs can determine changes to different dimensions of territorial capital, for example, strengthening community empowerment, bridging the social divide, developing fair and sustainable food chains, encouraging environmental protection, thus promoting human and ecological sustainability.

Studies of SI and rural development [15–20] underline the high potential of SI to improve the well-being of rural communities and societies, and the transition toward sustainability [15,21,22]. Reflections have been made on the importance of SI as a policy instrument to simultaneously create social benefits and economic opportunities [23–25].

Although SI has encountered support and resonance in many fields, two gaps remain to be filled. First, there is a lack of comparable systematic empirical evidence of the different social, economic, environmental, and institutional impacts that SI initiatives have at a local level. Research on SI is specifically lacking when dealing with sectors such as agriculture and forestry, and in marginalized contexts. Second, systematic tools for the measurement of the different impacts that SIs might have at a local level are missing [26]. For the characteristics of SI, it is necessary to use qualitative empirical evidence and quantitative indicators in a multidisciplinary approach [27,28]. Since SI is increasingly considered by governments, civil society, the private sector, and academia to offer solutions to complex problems, it becomes important to analytically understand, assess, and compare the various impacts of SI. A systematic assessment is important to assist policy makers and practitioners in designing, implementing, and supporting appropriate investment decisions, funding schemes, programs, projects, and policy design [29–31].

In this article we extend the work done in the EU Horizon 2020 project, Social Innovation in Marginalized Rural Areas (SIMRA), which developed analytical tools to understand SI and evaluate its impacts in European and southern Mediterranean marginalized areas [28,32] by means of a set of case studies related to the fields of agriculture, fishery, forestry, and rural development [33–35]. By employing the evaluation framework [27,32], we perform a systematic assessment of the impacts that nine SI initiatives might have at

a local level (i.e., administrative boundaries). The purpose is to answer the following research questions: (a) what impacts can SI initiatives have in marginalized communities and societies? (b) how can SI initiatives be clustered around specific combinations of impacts?

The article is organized into the following sections. We begin by clarifying the concept of SI, the SI impacts and their related aspects (Section 2); we present the framework used in this paper for the assessment of the impacts of SI initiatives (Section 3); we introduce the methods for the analysis of the data and the empirical material (Section 4); we present the results of the assessment (Section 5) and discuss the findings (Section 6); then we provide some concluding remarks (Section 7).

## 2. Theoretical Background

### 2.1. The Definition of Social Innovation

Since the 2000s, the concept of SI has been widely explored in the literature, resulting in many definitions and conceptualizations of SI [16,20,36–40]. They commonly refer to processes of the reconfiguration and transformation of networks, attitudes, and governance arrangements, aiming to address needs and improve the well-being of society. Building on the existing literature, in this paper we adopt the definition developed in the SIMRA project specifically referring to marginalized areas. The definition refers to SI as "*[ . . . ] the reconfiguring of social practices, in response to societal challenges, which seeks to enhance outcomes on societal well-being and necessarily includes the engagement of civil society actors*" [41,42]. SI is conceived as a: (1) process, (2) product, and (3) outcome [43]. First, SI is a process of social change, i.e., attitudes, practices, or perceptions [20,44–46]. Second, the product of SI refers to new formal and informal institutions, new and improved means of collaborative action, and new governance arrangements resulting from the process of reconfiguration, i.e., changes in social and institutional practices [47,48]. Third, SI is expected to boost outcomes and impacts on societal well-being [36]. Impacts and outcomes belong to the same category of effects of the SI but for different groups: while outcomes affect direct beneficiaries, impacts have effects on both direct and indirect beneficiaries. SI's final aim is to generate impacts on societal well-being. Based on this definition of SI, the following sub-Section 2.2 focuses on explaining different categories of impacts generated by SI initiatives.

### 2.2. The Impacts of Social Innovation

Impacts are long-term and widespread effects resulting from intentional and unintentional changes, which are determined by an accumulation of SI outcomes [49–51]. The impacts that SI initiatives may produce can be very different in terms of type, affected domains and scale. In the following section we discuss these three aspects in more detail.

#### 2.2.1. Types of Impact

SI emerges along a trajectory process. They may materialize in a tangible form (e.g., improved service offer) or in an intangible form (e.g., improved visibility and reputation on the national level). Its impacts can be positive, negative, or neutral [48].

Positive impacts lead to concrete benefits for the communities in which the SI has taken place [52]: they can be related to the reduction of different forms of marginalization [6,7], to the promotion of sustainable development, and at targeting sustainability challenges [10,53]. As rural marginalization is "*part of a broader process of social change, affecting society at large, and not particular to marginal localities [per se]*" [16], SI can have several positive effects on social, cultural, and human capital (e.g., increase social relations, boost self-organization and resilience, improve skills) [42,54]. SI can also lead to unintended negative impacts [49,55]. While the expected impacts of a SI should be positive, potential trade-offs in the community may arise as side-effects. SI may have positive effects at individual, community, and societal levels, or have internal impacts which are positive for the single SI (e.g., high value added of its services), but negative for external actors (e.g., competitor business-as-usual companies providing the same services). It may encourage the improvement of welfare services and society more generally, not necessarily beneficial

to all actors involved in SI [56]. It can also be a source of socio-political conflict, oppression, disempowerment of public structures, arising in response to changes in institutional logics, power relations and power distribution between and within different sectors and among stakeholders [57]. It can increase the sense of responsibility and social pressure in the actors responsible for the SI activities, that could feel "overloaded" by the work to be carried out [58]. This "overloading" is linked to the political expectation towards SI as a replacement for publicly funded services and as a" solution" for budget cuts [57]. Finally, an absence of impacts of SI initiatives, as well as unintended impacts (both negative and positive), can occur and should be considered [49].

### 2.2.2. Domains of Impacts

Depending on the type, dimension and location of the SI, impacts can be significant to environmental, social, economic, or governance/institutional aspects. For this reason, the impacts of SI can be identified according to domains, such as the society/community, economy, environment, and governance/institutions. The social domain of impact refers to social changes in terms of living conditions, health and well-being, human and social rights, aspirations and hopes, networks in the community, and cultural transformations [59,60]. The economic domain of impact refers to any change in the economy resulting from activities related to business opportunities, use of resources and the conditions for maximization of well-being determined by SI [61,62]. The environmental domain of impact refers to effects on the "surroundings in which SI operates, including land, water, air, natural resources, flora, fauna, humans, and their relationships" [63]. The institutional /governance domain of impact refers to changes in the coordination and decision-making processes amongst actors, including public administrations and policy makers, the private sector, and civil society actors, set off by the SI [64]. In aiming to achieve overall societal well-being, SI initiatives are expected to determine impacts on more than one domain, with various combinations. For example, social impacts could improve regional trust and loyalty that in turn can affect economy, governance, and environment. Similarly, environmental changes can determine social impacts, since the quality of the environment affects the life quality and livelihood of the community members. Even if it makes sense to consider SIs separated in relation to their social, economic, institutional, and environmental impacts, the separation of these categories is only an analyzing method to reduce complexity as there are many interdependences between them, as in the case of the Sustainable Development Goals [65].

### 2.2.3. Scale of Impacts

The scale of SI impacts can be spatial and social. Considering the spatial scale [66], impacts of SI initiatives can be evident inside the territory in which the initiatives take place, e.g., a municipality, a valley, province, or a region, depending on the type of challenge/need addressed and SI projects developed. Alternatively, SI initiatives can also have effects outside of the territory where they arise, i.e., at a provincial, regional, sub-national, EU or global level. The latter type of impact is difficult to achieve as SI is generally locally embedded, produced and connected to specific local projects and challenges [67,68]. Considering the social scale, SI initiatives can have positive impacts at micro, meso and macro levels [69]. They can have effects on the community (e.g., by providing social services), affect the whole society (e.g., fighting challenges of climate change) or it can address only the needs of some segments of the population and have effects at an individual level (e.g., empowerment of vulnerable groups such as women). For clarity, "community" is a common sociological term to denote a homogeneous group of people living together in one particular area, or people who are considered as a unit because of their common interests, while "society" is intended as a large and heterogenous group of people who live together in an organized way [70]. In this paper, we use "community" in both senses.

The scale, together with the domains and types of impacts are important aspects to be considered in the assessment framework of SI impacts, which will be presented in the next section.

## 3. Assessment Framework

The assessment of the impacts of SI aims to answer questions of cause-and-effect [50] and to identify changes that are directly attributable to a SI initiative, both intended and unintended. Different methods have been proposed to assess the impacts of SI initiatives [71–75], however, without producing commonly established impact measurements. This section briefly presents the evaluation framework developed by Secco et al. [32,52] to assess the impacts of the selected SI initiatives. The framework has been co-developed with stakeholders [27] and empirically tested in a set of case studies within the SIMRA project [76]. The assessment framework incorporates elements from the Theory of Change (ToC), applying it to a before–after comparison [51,77]. The ToC is typically used for designing, monitoring, and evaluating project interventions and their impacts. Applied to SI, the ToC approach describes the processes of change by outlining a linear causal relation between a SI initiative and its effects. The ToC is based on a results chain model organized according to a causal sequence that begins with SI inputs, moves through SI activities, its immediate results (outputs), and culminates in SI outcomes and impacts [77,78]. Both outcomes and impacts belong to the effects of a SI. However, while outcomes are mid-term effects on direct beneficiaries of the SI, impacts also have effects on indirect beneficiaries and become visible in the long-term. As a result, outcomes induce changes of practices (e.g., producing new routines), while impacts produce broader changes at a societal level (e.g., enhances civil society's capacity to act by empowering beneficiaries) [32].

Figure 1 shows a simplified depiction of the key elements of the ToC as applied in the SI evaluation framework [32], where the arrows represent the cause–effect relationships. Considering that SI initiatives typically include a process of innovation that leads to concrete socially relevant project(s), various elements have to be included in the evaluation [32]. First, since SI is a process of reconfiguring social practices, the changes in terms of new governance arrangements (e.g., institutional reforms), new networks (e.g., changes in structure or composition of actor networks) and new attitudes and values (e.g., cultural values, beliefs) have to be analyzed. Second, it is necessary to understand the social needs and/or societal challenges that SI initiatives seek to address (e.g., climate change, urbanization, etc.) and the context (i.e., material, and immaterial sets of resources) in which SIs arise and develop. These aspects enable us to better understand the magnitude of SI impacts at a territorial level. Third, SI is characterized by different actors, who join the process at different moments and generate different effects: the innovators are those who have the initial idea and start the initiative, i.e., the process of innovation (change); the followers join in the initial phases because they appreciate the innovators' ideas and want to contribute; the project partners typically enter into the process in a more advanced phase, enlarging the SI network (network members) and starting the practical activities for implementing a concrete project that determines outputs and outcomes; beneficiaries are those who get the benefits of using the services/products generated by the project, i.e., from the implementation of the SI project). Other stakeholders in the territory might also be affected by the SI initiative, both positively and negatively, even if they are not directly connected to it. As the actors are the core drivers of any SI initiative, an assessment of their perception is relevant [27]. Finally, SI generates immediate outputs from project activities, which in a later stage produce different types of effects on societal well-being (outcomes/impacts).

In this paper we measure the impacts of nine SI initiatives in marginalized communities by means of eight indicators specifically designed and applied for this purpose (see Section 4.2). Because of this concrete conceptualization of impacts, in the "before–after" comparison approach adopted in this paper, the identification and measurement of changes (i.e., effects or impacts of the SI initiative) are based upon perceptions of stakeholders directly or indirectly involved in SI. This will be further explained in the next section on methods.

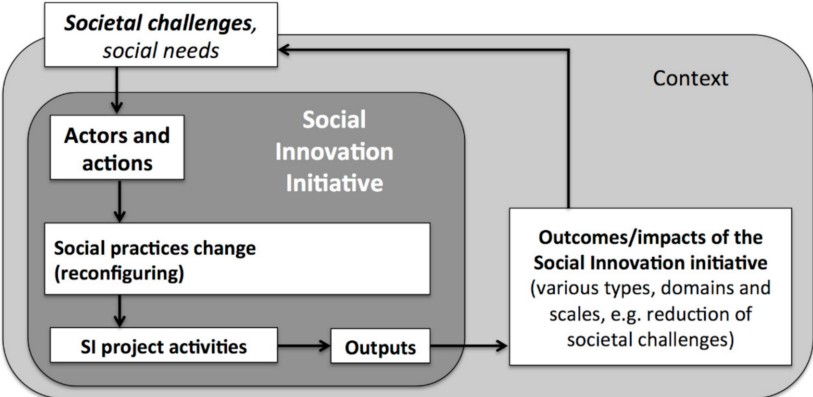

**Figure 1.** Applying the Theory of Change in the evaluation of social innovation (SI) and its impacts (source: authors, based on Secco et al. [32]).

## 4. Methods and Material

### 4.1. SI Initiatives under Examination

Nine SI initiatives were selected (Table 1; Figure 2) from the set of case studies of the SIMRA project on which a full evaluation (impact assessment) has been carried out [33,35]. These nine SIs exemplify a variety of initiatives in the field of agriculture, fisheries, forestry, and rural development, and represent the polymorphism of SI in European and southern Mediterranean areas characterized by different forms of marginality [6] and contexts [34,35]. Due to their diversity, these SIs are suitable for revealing a range of impacts of SI from an evaluation perspective (see [32]).

**Table 1.** Background information on the nine SIs (Source: Górriz-Mifsud et al. [33]).

| ID | Name of the SI Initiative with Brief Description | Location | Sector and Focus | Type of Marginality | Duration |
|---|---|---|---|---|---|
| WOM | Learning Growing Living with Women Farmers is a social cooperative offering delocalized childcare service provision on the farms. | South Tyrol (Italy) | Agriculture and Fishery: social farming | Socio-economic | 2006–Present |
| CAR | Green Care Farm is a private company handling the rearrangement of a farm into a green care farm for socio–economic integration of people with disabilities. | Walcheren Peninsula (The Netherlands) | Agriculture and Fishery: community agriculture | Physical constraints; limited infrastructures | 2003–Present |
| VAZ | VàZapp is a rural hub association promoting career opportunities for young people seeking to rejuvenate farmer entrepreneurship and create new business models. | Foggia (Italy) | Agriculture and Fishery: networking | Socio-economic | 2010–Present |
| DAI | Dairy Producer is a public–private partnership transferring extension and milk control activities from national institutions to Producers Organizations. | El Jem, Hazeg and Beni Hassen, (Tunisia) | Agriculture and Fishery: networking | Socio-economic | 2010–Present |

**Table 1.** *Cont.*

| ID | Name of the SI Initiative with Brief Description | Location | Sector and Focus | Type of Marginality | Duration |
|---|---|---|---|---|---|
| SEA | A Box of Sea is a project concerning the creation of a fairer market that protects the marine environment, supports small fishing communities, and provides better information to consumers regarding seafood. | Leros and Lesvos (Greece) | Agriculture and Fishery | Physical constraints; socio-economic | 2016–Present |
| FIN | Noidanlukko Cooperative is a grassroots network and platform aiming to document, communicate, and share knowledge on environmental issues and human–nature relationships, especially concerning nuclear power and other mega-projects planned in the area. | Pyhäjoki (Finland) | Rural Development:local development | Opponents of nuclear power marginalized in public discourses and policy. | 2017–2019 |
| LUM | Pro Val Lumnezia is a public–private partnership between young entrepreneurs, municipalities, and an environmental NGO created to stabilize a declining rural Alpine valley towards a sustainable tourism. In a reorganized form as an association, it has begun to cooperate with second homeowners on a regional level (Surselva region). | Val Lumnezia, Grisons (Switzerland) | Rural Development: local development | Socio-economic | 1986–Present |
| FLE | Réseau Urbain Neuchâtelois (Urban Network Neuchâtel) is a public–private partnership based on a regional contract established in 2007. It aims to strengthen the connection and cohesion within the Jura region as well as maintaining and developing its manufacturing profile. | Canton of Neuchâtel, (Switzerland) | Rural Development:urban–rural linkages | Socio-economic; inter-cantonal disparities | 2007–Present |
| LAG | Lochcarron Community Development Company is a nonprofit organization about community involvement in decision-making processes on woodland ownership, known as Community Forestry, and its strategic management for local development. | Strathcarron, (Scotland, UK) | Forestry: forest management | Physical constraints; limited infrastructures | 2009–Present |

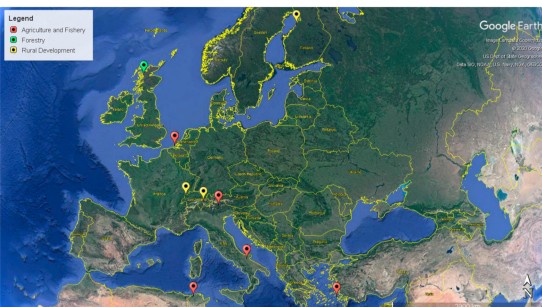

**Figure 2.** The location of the nine SI initiatives. © Google Earth.

*4.2. Data Collection and Analysis*

The empirical material was collected from November 2017 to September 2018 following the data collection and analysis tools designed by Secco et al. [52] (later refined in Secco et al. [32]). The data collection was organized into three phases and targeted to different types of stakeholders (i.e., respondents to interviews/questionnaires), divided into the SI categories of actors. These categories include the innovator(s) (i.e., the core group), the follower(s), the project partners, and the beneficiaries. Moreover, policy makers and external experts were interviewed as indirectly interested actors or with special knowledge in the SI initiative. Table 2 reports the number of respondents per category and SI initiative.

In the first phase of data collection, a focus group was held with the actors and key informants involved in the main phases of the SI. The key informants are the innovator(s) and follower(s), project partners, and policy experts and external actors with in-depth knowledge of the SI. Table 2 reports the number of participants who attended the focus group in each case study. Each focus group aimed to co-construct, with the participants, the storyline for the specific SI under analysis and collectively evaluate the perceived impacts of the SI in the four domains (social, economic, environmental, institutional). In the second phase, face-to-face semi-structured interviews were conducted with the core group involved in the SI, as well as with policy makers and external experts in each SI initiative, to collect the perspectives and interpretations of the actors on the impacts of SI across different domains within and outside the territory. In the third phase, structured interviews composed of open and closed questions were undertaken with the core group, network members, project partners and a sample of direct beneficiaries, to quantitatively understand the level of impacts that a SI has attained. The assessment of the impacts was mainly based on the actors' perceptions, both collective and individual, on whether and in which way the SI had/is positively or/and negatively impacted the social, economic, environmental, and institutional aspects of well-being. Thus, the perceived impacts are the subjective perceptions of the key knowledgeable respondents. This choice implies accepting perceptions as reliable "soft data", based on the assumptions that "all firms and individuals take actions based on their perceptions", that sometimes "it is difficult to come up with alternatives to perceptions data" [3,79], and that stakeholders highlighted the importance of perception in SI evaluation [27]. Therefore, due to different lifespans of the SI initiatives (see Table 2 for information on the durations of the SIs), the assessment of perceived impacts might reflect: (a) already attained effects (i.e., the Women Farmer Cooperative (WOM), Green Care Farm (CAR), Dairy Producer (DAI), Urban Network Neuchâtel (FLE), Pro Val Lumnezia (LUM)), (b) effects that communities are receiving presently or will receive in the near future (i.e., Lochcarron Community Development Company (LAG), A Box of Sea (SEA)), or (c) effects that may take place over the long-term (i.e., Noidanlukko Cooperative (FIN)). Since the FIN case was initiated only two months before the assessment has been conducted, the impacts measured correspond to potential future impacts.

**Table 2.** Number of focus groups, semi-structured and structured interviews performed in each SI initiative by respondent type (source: authors).

| SI Initiative | Focus Group Participants | Semi-Structured Interviews | | Structured Interviews | | | |
|---|---|---|---|---|---|---|---|
| | | Core Group | Policy Makers and External Experts | Core Group | Network Members | Project Partners | Beneficiaries |
| WOM | 4 | 1 | 3 | 2 | 4 | 8 | 7 |
| SEA | 10 | 2 | 0 | 2 | 12 | 4 | 9 |
| VAZ | 10 | 3 | 3 | 3 | 25 | 0 | 10 |
| LUM | 7 | 4 | 3 | 1 | 10 | 1 | 6 |
| FIN | 9 | 6 | 3 | 1 | 2 | 1 | 6 |
| CAR | 7 | 4 | 2 | 1 | 3 | 3 | 16 |
| DAI | 9 | 10 | 0 | 2 | 6 | 1 | 1 |
| LAG | 5 | 2 | 5 | 1 | 1 | 3 | 5 |
| FLE | 5 | 2 | 1 | 2 | 4 | 2 | 2 |
| TOTAL | 66 | 44 | 24 | 19 | 67 | 23 | 62 |

Data collected were used to build and calculate quantitative indicators that evaluate the SI impacts according to the methodology and formulae developed by Secco et al. [32] and explained in detail in the paper by Pisani et al. [28] in this Special Issue. The eight indicators of impacts (Table 3) used in this study can be grouped into three main analytical dimensions: type of impacts, domains of impacts, scale of impacts. The indicators refer to the three aspects of the impacts described in Section 2.2. In terms of type of impacts, the indicators applied (the codes of which are identified in brackets) measure the proportion of marginalization problems improved by the SI initiative (A1) and the level of improvement in European Union societal challenges due to the SI initiative (A2). In terms of domains affected by the impacts, the indicators applied measure the balance of positive to negative significant impacts of the SI initiative in the four domains (economy, society, environment and institutions). This measurement is according to the perception of stakeholders (B1), the proportion of the number of impacts of the SI initiative in the four domains which were positive, according to the stakeholders (B2), the level of effects of the SI initiative in the four domains according to the actors (B3), and the level of improvement in governance aspects due to the SI initiative, according to the perception of the actors (B4). Finally, in terms of the scale of impacts, the indicators investigate the level of effects of the SI initiative inside (C1) and outside the territory (C2) in the four domains according to the perceptions of the actors.

The indicators are based on a "before–after" comparison, as no counterfactual analysis could be performed [32]. Due to the nature of the information collected, indicators use different ranges (e.g., 1–10; 0–100); hence to make them comparable they have been normalized in the 0 to 1 range. Different normalization approaches have been tested (Min–Max with respect to the data collected, categorical scales, and indicators above or below the mean). However, the relatively small number of cases on which they have been calculated did not permit checks on their reliability and generalization. For this reason, the Min–Max (with respect to the indicator range) normalization approach has been adopted, except for the indicator B1. According to the indicators developed by Secco et al. [32] the upper limit of B1 is infinite. This value is obtained during the focus group whenever no effects have been indicated as negative by participants. However, in this study, the infinite value was replaced with the sum of the positive effects, to differentiate the case studies and allow normalization, as it was done for the other indicators. As a result, in our study the range of the indicator changed from (0–inf) to (0–1024) and then normalized to the (0–1) range.

**Table 3.** List of indicators selected for the assessment of the impacts of the SI initiatives (Source: Secco et al. [27,52]). For a detailed explanation of the indicators see Appendix A, Table A2.

| Category of Impacts | Name and Meaning of Indicator | | Respondent Type |
|---|---|---|---|
| **A.** **Type of Impact** | A1 | Proportion of marginalization problems improved by the SI initiative, as perceived by stakeholders | Stakeholders taking part in the focus group |
| | A2 | Level of improvement in European societal challenges due to the SI initiative, according to actors. | Core group; project partners |
| **B.** **Domains of Impacts** | B1 | Balance between positive to negative significant impacts due to the SI initiative in the four domains according to the perceptions of stakeholders | Stakeholders taking part in the focus group |
| | B2 | Proportion of the number of impacts of the SI initiative in the four domains which were positive, according to the stakeholders | Stakeholders taking part in the focus group |
| | B2_soc | Social domain e.g., life satisfaction and happiness, solidarity and mutual trust among the members of the community, civic engagement | |
| | B2_eco | Economic domain e.g., household income; employment opportunities and quality; labor conditions in the sector in the territory | |
| | B2_env | Environmental domain e.g., pollutant emissions to air; landscape and ecosystems; biodiversity | |
| | B2_ins | Institutional domain e.g., stakeholder empowerment and representativeness in decision-making process; capability of public administrations to manage collaboration | |
| | B3 | Level of effects of the SI initiative in the four domains, according to the actors | Core group; project partners; beneficiaries |
| | B3_soc | Same as B2_soc | |
| | B3_eco | Same as B2_eco | |
| | B3_env | Same as B2_env | |
| | B3_ins | Same as B2_ins | |
| | B4 | Level of improvement in governance aspects due to the SI initiative, according to the actors | Core group; project partners |
| **C.** **Scale of Impacts** | C1 | Level of effects of the SI initiative inside the territory in the four domains, according to the actors | Core group; project partners; beneficiaries. |
| | C2 | Level of effects of the SI initiative outside the territory in the four domains, according to the actors | Core group; project partners; beneficiaries. |

The results of the calculation of the indicators of impact in each SI were analyzed by using both qualitative (i.e., qualitative analysis of interviews) and quantitative tools (i.e., univariate, and multivariate statistical techniques). To come to the results, the quantitative analyses of the indicators are integrated and complemented by the triangulation with

qualitative information extracted from the interviews, the focus groups, and the open-ended questions in questionnaires. The results are structured in three phases of analysis.

First (Section 6.1), we used descriptive univariate statistics to describe in a simple way the main impacts of the nine SI initiatives by triangulating qualitative and quantitative information [33] following the approach proposed by Secco et al. [32]. Qualitative data are integrated, which contribute to a better interpretation of the quantitative values of the indicators and to better understand the different aspects of impacts in each SI initiative. Second (Section 6.2), we employed multivariate statistical methods to understand how mutually dependent indicators behave collectively and to explore if SI initiatives can be grouped together. The correlation matrix was used to identify existing dependencies across indicators showing how they tend to vary together, positively, or negatively (Figure 3; Figure 4). A Pearson correlation matrix distance was computed using all the indicators of impacts presented, including the sub-dimensions for indicators B2 and B3. Based on the results of the correlation matrix, a principal component analysis (PCA) was run to visualize generic distance and relatedness (cluster) between variables in a three-dimensional space and to capture their variability [80]. It was also used to summarize multivariate datasets into a few principal components that are expected to account for most of the variability, and to group indicators according to certain categories of impacts. A PCA was computed using all the impact indicators employed in the correlation analysis and the social, economic, environmental, and institutional domains of the indicators B2 and B3. The results of the analysis are visualized in two-dimensional scatter plots (Figure 5), which organize the dataset according to pairs of principal components. In the scatter plot, the observations (i.e., the SI initiatives) sharing the characteristics explained by the principal components are expected to be spatially located near each other. This enables the identification of whether the SI initiatives cluster around specific combinations of the impact indicators. Additionally, a PCA enables the identification of whether the impact indicators work synergistically or not and can distinguish groups of individual indicators that are responsible for the majority of the variations/according to their degree of correlation with the principal components. Multivariate analyses were performed using the statistical software R [81].

## 5. Results

### 5.1. Main Impacts of the Nine SI Initiatives

This sub-section presents the impacts that each SI analyzed has created or is expected to create on its territory, focusing on the most relevant aspects according to qualitative information and quantitative values of the indicators (see Appendix A, Table A1). The values of the indicators are interpreted as follows: the higher the values, the higher the performance of the SI with respect to the indicators. SI initiatives are presented here according to the(ir) main sector of intervention: agriculture and fishery (SI initiatives WOM; CAR; VAZ, DAI; SEA), rural development (LUM; FLE; FIN) and forestry (LAG) (See Table 1 in Section 4).

The Women Farmer Cooperative (WOM) in Italy responded to the need for the provision of social and care services in remote mountain valleys, and to the need for an increase in the number of professional opportunities for women farmers. The initiative's positive social impact is the transformation of the farm into an educational place and meeting point for families in the community. This social impact is confirmed by high values of both the indicators measuring the proportion of the number of social impacts in the SI initiative which were positive (B2_soc = 1) and the level of effects on the social domain inside and outside the territory (B3_soc = 0.87). By providing a care service in the remote mountain valleys of the region and offering qualified working opportunities, the initiative has improved overall social cohesion in the wider social territorial system, as confirmed by a high value of the indicator regarding the impact on governance aspects (B4 = 0.88), and about the ability to address societal challenges (A2 = 0.82).

The cooperative Green Care on the Farm (CAR) in the Netherlands responded to the need to integrate people with disabilities in the labor market in a rural, remote area.

Through the initiative and despite initial difficulties, the development of the care service on the farm has determined the start of an intensive collaboration amongst farm owners, volunteers, local government, and a regional organization of green care farms. The initiative has developed into a new business model for delivering tailor-made care services that diversifies agricultural income and determines economic development in this marginalized rural area. The positive impacts in terms of a reduction of marginalization and on the socio-economic and governance aspects are demonstrated by high values of related indicators A1, B2_soc, B2_eco, B2_ins, B4. The initiative has contributed to addressing societal challenges, as shown by the high value of indicator A2 (0.90).

The creation of the farmers' network Vazapp (VAZ) in Italy generated impacts mainly on the social domain (B2_soc = 1 and B3_soc = 0.80), creating a positive image about local and traditional agriculture, portraying it as a sector with development opportunities for young entrepreneurs. Stakeholders clearly perceived the initiative as a "soft device" to facilitate social integration into the agricultural sector. Stakeholders perceived the high positive impacts of the SI in all domains (B2 = 1). The impacts generated by Vazapp were visible both inside and outside the territory. Inside the territory, young farmers involved in the initiative have increased their level of self-confidence, besides improving the image about their own territory (C2 = 0.70); outside of the territory the initiative has been able to inspire policy makers in defining new integrated development strategies in the agri-food sector.

The private–public partnership of dairy producer organizations and national institutions (DAI) in Tunisia has been successful in improving relationships and cooperation between the two actors. The initiative started from the government's need to improve dairy productivity in Tunisia. Through the support of FAO (Food and Agriculture Organization) in designing a horizontal process, impacts were observed in the improvement of social and economic conditions of dairy farmers, and of the areas where they are located in general, as shown by a high value of B2 (0.97). This has been achieved through the improvement of sustainable income-generating activities and by enhancing food security and social inclusion as shown by the high value of A2 (0.88).

The Box of the Sea initiative (SEA) in Greece started from the idea of developing and showcasing a financially and environmentally sustainable small-scale business model in the fishing industry. As a result, the targeted impacts were mainly social and economic, but also environmental, and outside the territory, as shown by the high values of indicators B3_soc (1), B3_eco (0.86), B3_env (0.78) and C2 (0.75). The impacts of the initiative were mainly concerning informing consumers who participate in the project, often living in urban areas, about sustainable fishing and consumption practices, as well as creating a producer–consumer relationship based on the delivery of freshly and sustainably fished sea products. The high values in B3_eco and B3_soc reflect the impacts of the project on the (fishers') market position of actors, stabilizing their income and employment, and on the development of their cooperative spirit, which was previously lacking.

The cooperative Noidanlukko (FIN) in Finland aims to provide a forum for the discussion of key environmental and social challenges through its Information Center, Hanhikivi. Social challenges emerged regarding the nuclear power plant project at Pyhäjoki, Finland. Already in the planning phase, the nuclear project has led to the social and political marginalization of opponents of nuclear power and has divided the local community. The proponents expect major economic benefits from the nuclear power plant project while the opponents fight against perceived injustice (e.g., land acquisition by force; risks of negative environmental impacts; lack of participation opportunities). The social innovation actors (i.e., members of Noidanlukko cooperative and opponents of nuclear power) perceived that the initiative could have a high ability to reduce the social marginalization of the territory (B2_Soc = 0.90). The SI actors perceive future impacts on the environment as potentially huge (B2_env = 1.00) in that they could be able to stop the nuclear project and thus avoid (possible) negative impacts.

The Pro Val Lumnezia (LUM) initiative in Switzerland emerged from the need for young craftsmen trained outside the valley to initiate structural change in an agriculturally dominated and demographically declining Alpine valley. To increase the attractiveness of their region in a sustainable manner, the actors cooperated with regional consultants and with a nature conservation organization. The initiative succeeded in overcoming a condition of territorial fragmentation in terms of decision-making processes, and therefore in strengthening the role played by the valley within the larger canton of the Grisons. This is reflected in the high value of B4 (0.71) about improvement of governance arrangements, and a very high value of B2 (1) about the perceived positive impacts by the stakeholders.

The Réseau Urbain Neuchâtelois initiative (FLE) in Switzerland emerged from the need to increase cooperation amongst the cantonal administration of Jura and its valleys to secure the canton's industrial profile in the long term. The main outputs of this initiative have been the creation of regional network structures linking various Jura valleys with the canton's two largest towns, the merging of the small municipalities of the Travers valley (Val de Travers) into a single municipality, and the institutionalized cooperation amongst the main industrial companies in various sectors and the municipal and cantonal administrations. The resulting constellation and cooperation between public, economic and civil society institutions can be highlighted as the core impact of the SI. The improvement of governance arrangements by the initiative is confirmed by the high value of the indicator B4 (0.96).

The Community Forestry (LAG) in Scotland aimed to enable a community's involvement in the ownership and strategic management of the local woodland for local development. The initiative's positive environmental impacts can be observed in the increased environmental benefits brought about by the clearing of dead trees and the replanting of native coniferous tree species in the woodland, as highlighted by the value of indicators B2_env (1) and B3_env (0.77). Through the development of an access path to the woodland and a recreation trail, as well as by using the woodland for educational purposes, the initiative increased well-being and social benefits such as a sense of place, community cohesion and empowerment (B2_soc = 1 and B3_soc = 0.72). Although the SI has improved governance aspects in terms of community's involvement in the management of the woodland, it has had limited impact on public administrations.

The average values of the impact's indicators calculated for the nine SI initiative reveal some general trends. Indicator A2 shows a high average value across the nine initiatives, meaning that addressing societal challenges is at the core of SI. The average value of B1, highlighting the balance between positive and negative impacts, is quite low and positive, suggesting that SI might also have some negative impacts, which are counterbalanced by the positive ones. Indicator B2 shows high average values for the social, economic, governance domains (B2_soc, B2_eco, B2_ins). This means that the actors participating in the focus groups acknowledged high positive effects of the SI under investigation. Indicators assessing the environmental domain (B2_env) show lower values compared to other domains, meaning that SI initiatives have more impacts on the socio-economic and institutional aspects, while still aiming to achieve environmental sustainability. Finally, the initiatives generally had more impacts inside the territory than outside, as proven by the higher average value of C1 with respect to C2. The following sub-section reports on the average indicator values and on existing correlations amongst indicators.

### 5.2. Trends among Variables Measuring the Impacts of SI Initiatives

The correlation matrix shows the main relationships between the impact indicators (Figure 3). The range of values obtained varies from 0 to 1. This means that the eight variables measure different aspects of impact within the SI initiatives. The main relationships are reported below.

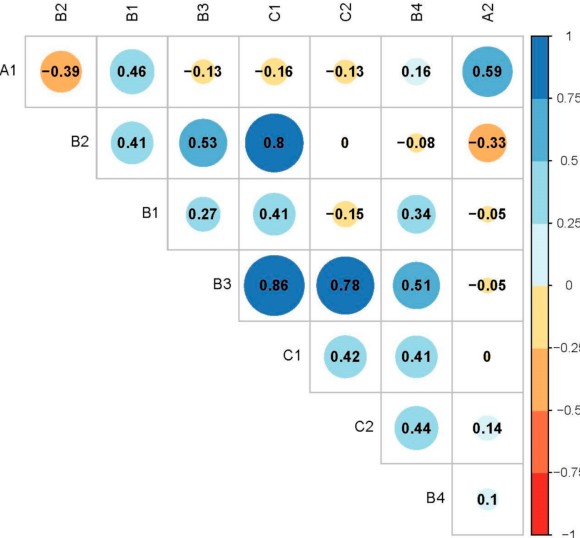

**Figure 3.** Correlation matrix of seven indicators of impacts of SI. Blue colors indicate a positive correlation between variables, while red–yellow colors indicate a negative correlation. The size of the bubble and the darkness of the color indicates stronger (positive or negative) correlation coefficients. Indicators are reported with their respective acronyms, a full description of the indicators is provided in Table 3. (Source: authors).

The strongest linear positive correlation is between indicators B3 and C1 (0.86). This means the actors perceived that the level of effects of the SIs in the four domains are mainly happening inside the territory where the initiatives develop. The strongest negative correlation is between A1 and B2 (−0.39) indicating that the stakeholders perceive that an increase in the number of positive impacts on the four domains does not necessarily correspond to a decrease in marginalization problems. By comparison., indicator A1 is highly positively correlated with indicator A2 (0.59), which means that stakeholders perceive there is a relationship of mutual dependence between the capacity of the initiatives to address problems of marginality and to resolve EU societal challenges.

A positive correlation was found between the indicators B3 and B4 (0.51), showing that when actors perceived the SI initiative to have achieved high impacts in the four domains, they also perceived the initiatives to have significantly improved governance arrangements, thus generating stronger impacts at the institutional level. Indicator B3 is also highly positively correlated with B2 (0.53), indicating that both SI actors and stakeholders perceived SIs as creating positive impacts, although their perceptions differ regarding the magnitude of the effects of the initiatives.

The correlation matrix performed amongst B2 and B3 can be split into four sub-indicators which show the impacts on the four domains (environmental, social, economic and governance). As shown in Figure 4, B2 is equally explained by the four domains, meaning that stakeholders perceive that SIs generate positive impacts on environmental, social, economic, and governance/institutional aspects. By comparison, indicator B3 is mainly explained by the economic and institutional domains. This means that the actors involved in the initiatives perceived that SIs generate higher impacts in only two domains. For both indicators B2 and B3, the economic domain is strongly positively correlated with the social domain (B3 = 0.72 and B2 = 0.9) and with the institutional domain (B2 = 0.92 and B3 = 0.55), meaning that the higher the economic impact achieved by the SI initiative, the higher its social and institutional impacts (and vice versa). Also, the social domain is highly correlated with the institutional one (B3 = 0.62 and B2 = 0.85). In both cases, the environmental domain has null or negative correlation with the other domains of the indicators.

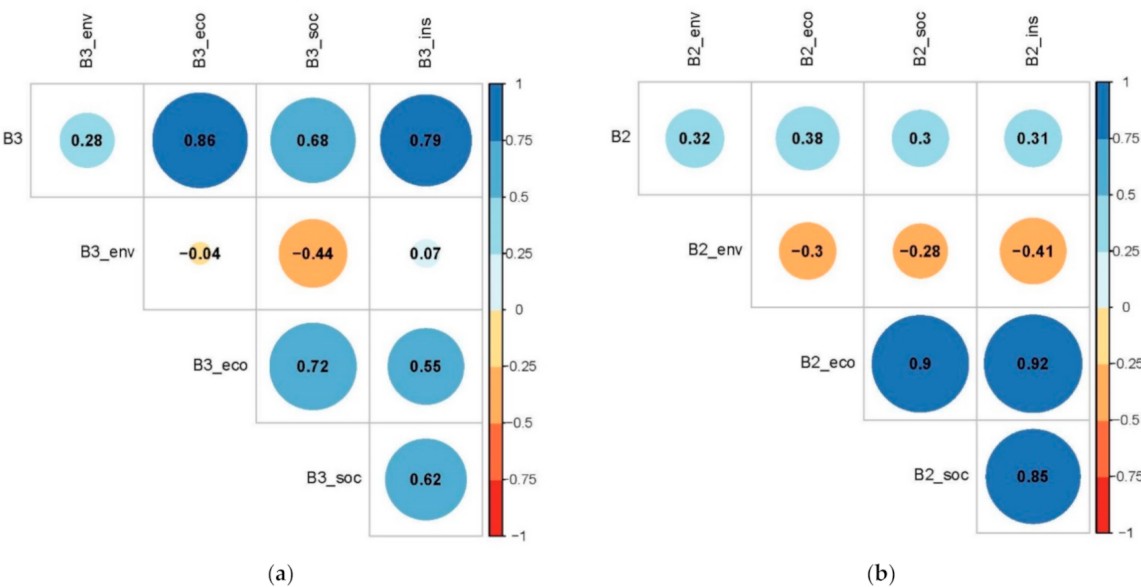

(a)                                                                 (b)

**Figure 4.** Correlation matrices of indicator B2 (**a**) and indicator B3 (**b**) and the respective four subdomains (environment, economy, society and institutions). Blue colors indicate a positive correlation between variables, while red–yellow colors indicate a negative correlation. The size of the bubble and the darkness of the color indicates stronger (positive or negative) correlation coefficients. Indicators are reported with their respective acronyms, a full description of the indicators is provided in Table 3. (Source: authors).

### 5.3. Trends among Variables Measuring the Impacts of SI Initiatives and Cases of SI

The results of the PCA show that three axes explain 69% of variability of the indicators. We interpreted the three axes as follows: Axis 1 refers to the impacts of SI initiatives on the four domains, Axis 2 is about the impacts of SI initiatives on marginalization and addressing EU societal challenges, while Axis 3 represents the magnitude of impacts of the SI initiative within the territory. In the following sections we explain them in detail.

#### 5.3.1. Axis 1: Impacts of SI Initiatives on the Four Domains

The first axis focuses on the impacts on the four domains (society, economy, environment, and governance) (Figure 5). The focus of the impacts is at a local level (C1). This axis is mainly explained by the environment domain on one side and by the social, economic, and institutional domains on the other side. The values of the indicators assessing environmental impacts were lower when compared to the other domains. Nevertheless, the environmental domain (analyzed by B2_env) is clearly counterpoised to the others (analyzed by institution, social, economic domains of B2 and B3, and B4). The cases that explain this axis are those that achieved environmental impacts (even though their values were low), and those achieving socio-economic and governance/institutional impacts. The Finnish cooperative is an example of the environmental impacts: it raised awareness of the negative effects of nuclear power. Similarly, the Scottish (UK) case on community forestry promoted the valorization of forest resources and a more sustainable management. These SIs show that environmental impacts of SI have been achieved in very different fields by re-organizing planning, management, and distribution processes collectively/cooperatively. There are other cases that have, primarily, impacted the institutional and economic domains. For example, the Tunisian private–public partnership aimed to enhance relationships between producer organizations and farmers whereas the Swiss public–private partnership aimed to increase the attractiveness of the area for manufacturing.

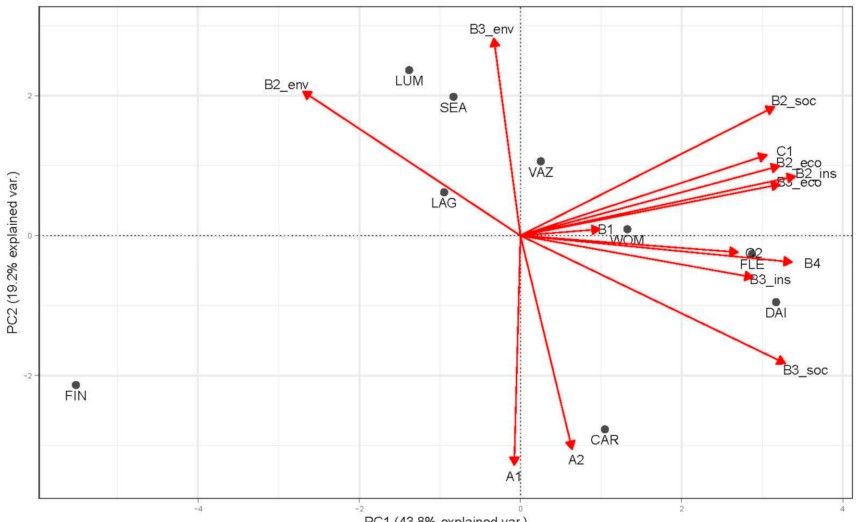

**Figure 5.** Principal Component Analysis (PCA) ordination diagram of sampled social innovation case studies (*N* = 9) and impact variables (*N* = 14). PCA axis 1 and 2 are displayed alongside the percentage of explained variance. The acronyms of the SI initiatives are provided in Table 1. Indicators are reported with their respective acronyms; a description of the indicators is provided in Table 3. (Source: authors).

5.3.2. Axis 2: Impacts of SI Initiatives on Marginalization and in Addressing EU Societal Challenges

The second axis focuses on marginalization (Figure 5 and Figure 6). Not all SI initiatives were able to create widespread impacts on the territory to address and decrease marginalization (A1), and also address some of the EU societal challenges (A2). The axis therefore contrasts the initiatives which do not focus on responding to these challenges with those that have managed to generate concrete impacts on marginalization. Amongst the latter group of initiatives are the Dutch initiative (DLO) and the Finnish cooperative (FIN). They represent the types of initiatives that recognize or respond to tackling marginalization and addressing various EU societal challenges.

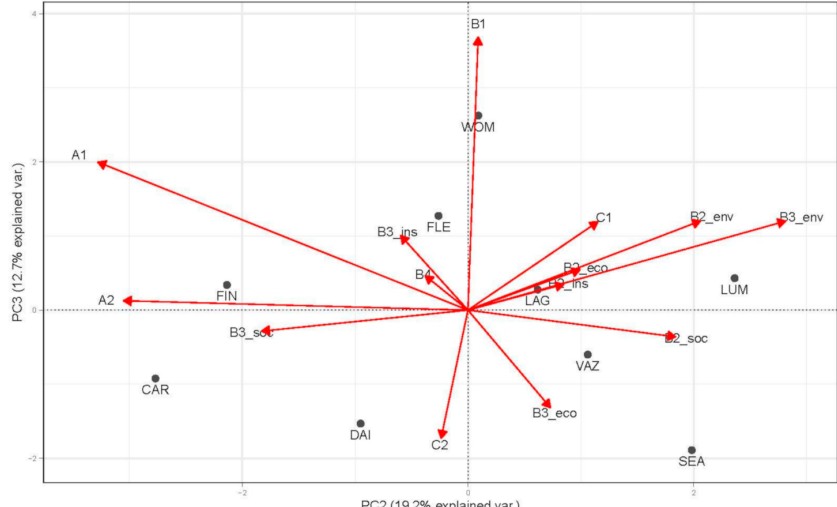

**Figure 6.** Principal Component Analysis (PCA) ordination diagram of sampled social innovation case studies (*N* = 9) and impact variables (*N* = 14). PCA axes 2 and 3 are displayed alongside the percentage of explained variance. The acronyms of the SI initiatives are provided in Table 1. Indicators are reported with their respective acronyms; a description of the indicators is provided in Table 3. (Source: authors).

The development of the green care farm in a rural, marginal area of the Netherlands meets the demand for social care provision for vulnerable groups, such as people with disabilities as well their socio-economic integration, focusing on the European well-known societal challenge of creating socially inclusive communities. The Finnish cooperative aims to create a network among local people, international experts, artists, and journalists in opposition to the nuclear power plant project. In relation to EU societal challenges, this case has provided seeds of reflection on alternative rural development strategies by challenging the narrow focus on technology-oriented economic growth as the only viable option to enhance social well-being.

5.3.3. Axis 3: Magnitude of Impacts of SI Initiatives within the Territory

The third axis focuses on the magnitude of impacts (Figure 6 and Figure 7). It is mainly explained by indicator B1, which studies the balance between positive and negative impacts in the four domains according to the actors, and by indicator C2, which measures the level of the effect of the SI outside the territory in the four domains. The axis contrasts cases in which the level of impact is mainly occurring locally with those cases in which the level of impact in the four domains occurs mainly outside the territory (C2). The women farmer cooperative (WOM), in Italy, and the Box of the Sea cooperative (SEA) supporting fishermen in Greece can be used as examples to explain this axis.

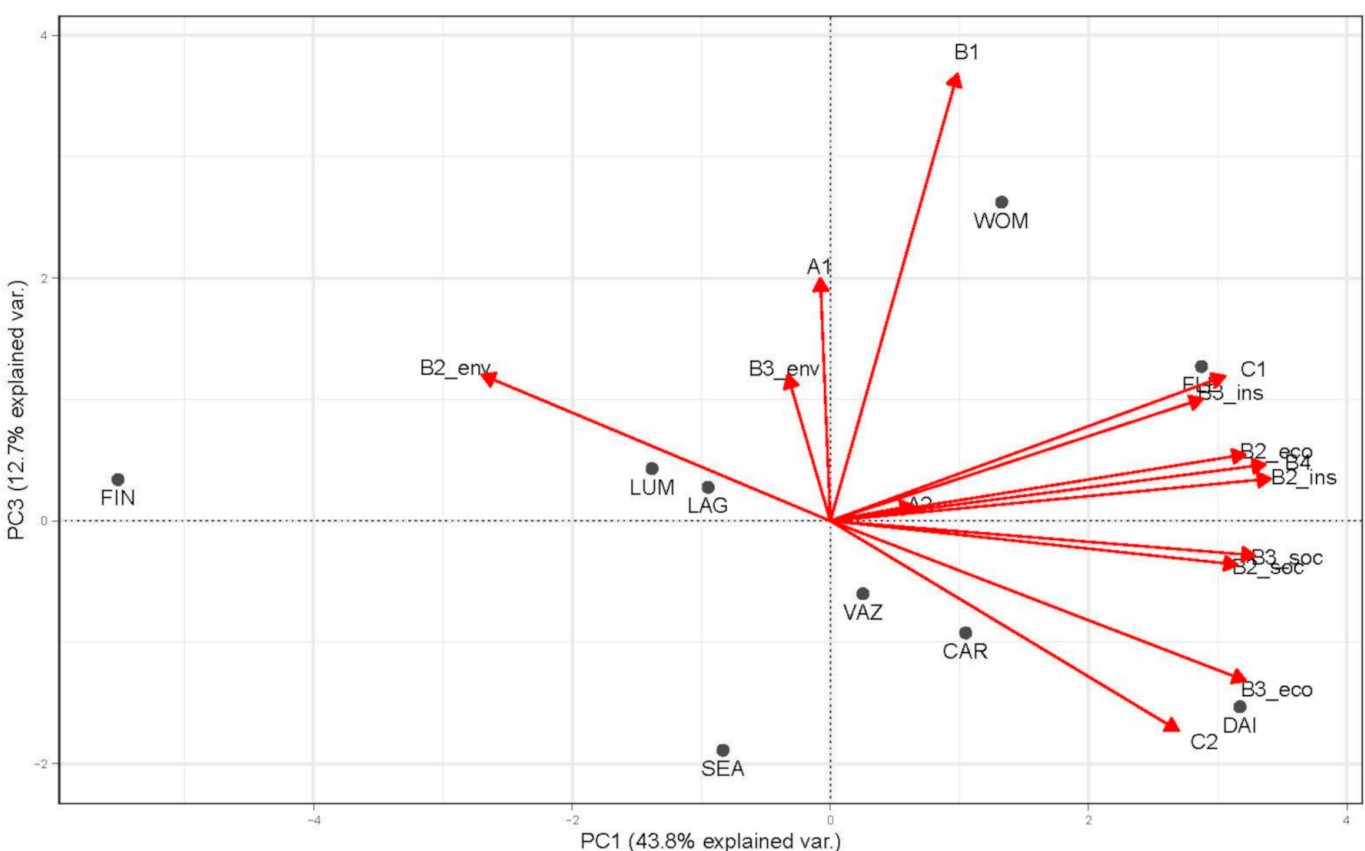

**Figure 7.** Principal Component Analysis (PCA) ordination diagram of sampled social innovation case studies (*N* = 9) and impact variables (*N* = 14). PCA axis 1 and 3 are displayed alongside the percentage of explained variance. The acronyms of the SI initiatives are provided in Table 1. Indicators are reported with their respective acronyms; a description of the indicators is provided in Table 3. (Source: authors).

The first initiative aimed to create on-farm childcare services for local families living in rural, remote villages in South Tyrol. The impact of the Italian initiative WOM has been the empowerment of women farmers willing to continue living in rural areas, while creating for themselves a professional role on the farm and gaining independence from the male

members of the family. Hence, the impacts were mainly visible in the improvement of the conditions of the rural communities in which the initiative emerged and developed, although the initiative has been considered as an inspirational model in other contexts. The Greek cooperative is an initiative aiming to create a fairer and innovative model of marketing and distributing seafood which protects the marine environment and supports small fishing communities in a cooperative model. By linking remote islands to the Greek capital, SEA delivered impacts outside the local community of Lesvos and Leros where the initiative is based.

## 6. Discussion

The analysis of the impacts of the nine SIs has shown an improvement in the overall conditions of the communities and societies in which the SIs emerged and developed. This section discusses the most relevant results of the analysis (Section 5) by contextualizing them within the existing literature.

### 6.1. Positive and Negative Effects of SI Iinitiatives

Our results show that, according to the perceptions of the actors and stakeholders, SI initiatives have generated or are generating impacts that are positive overall. In five out of the nine SI initiatives, it has been perceived that SI improved the socio-economic condition and cultural livability of the rural communities, promoting sustainability outcomes. SI provided business diversification, created new economic opportunities, promoted the empowerment of vulnerable groups, and enhanced cooperation and networks across sectors and actors, thus enhancing the social capital. In two out of nine cases, the effects of the SI initiatives have become visible even if the initiatives are quite recent, and in one case, the impacts are expected in the future, thus providing directions for planning and policy making. These empirical data are consistent with studies showing that promoting SI in rural and marginalized areas is important, as it has positive effects mainly on the society, economy, and governance (e.g., [17,19,58,76,82–88]). By taking the forms of social cooperatives, associations, social enterprises and private–public partnerships, SI proposes innovative solutions, models, services and approaches that tackle different complex social issues (e.g., labor market integration, social exclusion and poverty, discrimination, social service delivery, and disproportionate resource uses). These solutions are co-created with citizens; this enhances society's capacity to act, leading to an increase in social empowerment, social inclusion, and social capital, and creating the condition for changes in society's norms and values [82]. At the same time, SI develops by means of socially responsible businesses in certain sectors of the economy and proposes new economy models (e.g., social economy, collaborative economy) that support the creation of innovative, vibrant, and sustainable societies [15]. Finally, when SI is supported by strong and functioning institutional arrangements, which allows the involvement of non-traditional actors (local authorities) and a cross-sector collaboration (among government, markets, and civil society), it promotes innovation processes in politics and governance which are beneficial for the community in which SI occurs [84]. Nevertheless, it is also necessary to recognize that SIs may create trade-offs which are not always positive, and that do not affect SI actors in the same way. Indeed, SIs may simultaneously have empowering effects on some actors and disempowering effects on others, or empowering effects with some limitations as well as negative downturns [57]. Therefore, it is likely that coping better with marginality and EU societal challenges will simultaneously disempower some actors.

### 6.2. SI Reduced Certain Forms of Marginality While Addressing Current EU Societal Challenges and Territorial Disparities

Our results show that SI initiatives were able to both tackle problems of marginality and address some of the most pressing EU societal and territorial challenges. Marginalization is one aspect of growing spatial disparities and inequalities, and is embedded in broader processes of social change, affecting society at large, and is not specific to marginal localities [89,90]. Marginalization may occur in any region, even in those that are more

centrally located, if their former connections break or lose significance [16]. In the search for alternative pathways, SI initiatives mitigate these structural problems. Indeed, several elements (e.g., ageing of the population, brain drain, remoteness, etc.) that used to define marginalization and marginalized areas [6], are aligned with EU societal challenges (e.g., ageing of the population; income, jobs, education; inclusive societies; innovative societies; environment and climate change). Moreover, some of the EU policies for rural development, where SIs can be applied at a local level (e.g., smart villages, and the LEADER approach in the EU Rural Development Programme), are directly tackling some of the EU societal challenges [91]. Hence, in their approaches to tackling current European problems, SIs adopt creative and inspiring approaches to tackling European regional policies but have not yet achieved the systemic change which is the final goal of transformative SIs. These results are in line with the existing literature [4,5,57,86].

### 6.3. SI Generated Impacts Mainly within Their Local Territories

Our results show the local character of Sis, and that their effects mainly occur inside the territory in which the SI emerges and develops. This can be explained by the fact that SIs arise mainly in response to local problems and are embedded in specific socio-political and socio-economic contexts, being locally rooted in the territory where they develop and evolve [92]. SIs are driven by the activities of local actors. Scaling up SI initiatives is difficult, because innovators do not have the necessary economic and human resources (such as funding, volunteer time and skills within a community) to replicate the initiative beyond their local context [40]. Often, local actors are not interested in achieving impacts beyond their territory, or do not have the necessary networks outside of their territory to scale up their initiative, and as a result they mainly remain committed to their own community. Although these results do not show a trans-local character of SIs and its impacts, the literature reports that SIs can involve a broad network of actors (e.g., cooperation between people, organizations, and institutions) and reach beneficiaries more widely, thus having impacts that spread outside the local territory in which SIs emerge and develop [93].

### 6.4. SI impacted Mainly on the Social, Economic, and Institutional Domains

Our results reveal the cross-sectoral nature of SI and the interdependent character of the impacts that create chains of different types of effects. The SIs analyzed show a correlation between the social, economic, and institutional domains that simultaneously generate overall positive impacts on societal well-being. For example, the provision of services in response to new societal needs, market failures or austerity in public policies (social domain) determine innovations in governance and the reorganization of the welfare system [94]. SI, as a process of the reconfiguration of social practices, develops new partnerships, collaborations, networks, etc. based on trust, reciprocity, collaboration, and autonomy [57,84,95]. Our empirical results corroborate the literature on the effects of innovations in existing relationships, community ties, and new forms of collaboration to have positive effects on the modernization of public administration, on the reduction of bureaucracy, and on the level of efficiency in the delivery of services in the public sector (governance domains). Our results agree with the literature reporting that SI initiatives that make services available to communities and to specific vulnerable groups (social domain) can generate effects on the economic domain in different ways (e.g., increase employment opportunities, provide resources for generating other activities, and the development of human capital potential) [16,46,58,88,96]. Finally, the economic impacts may have effects on the social domains in terms of a reduction of inequalities and in the improvement of well-being, such as improving community relationships (e.g., intergenerational relationships, integration of vulnerable groups) [62,73]. Although our results do not show a clear synergy with the environmental domain, another recent study has identified SI contributions to the environmental aspects of the Sustainable Development Goals [91].

Overall, SI initiatives have helped societies and communities to sustainably cope with EU societal challenges and to further expand their capacity to achieve sustainability

ambitions (e.g., sustainable income-generating activities, sustainable use of forest resources, sustainable product production and food chains, food security). This is in line with the literature showing that SI further helps communities to move from current unsustainable models of living to new, sustainable ones, accompanying the social and economic transitions of societies [97].

### 6.5. SI Achieved Multi-Level Impacts

The results show that the SI initiatives analyzed have achieved impacts on three levels: the micro level of individuals, the meso level of community, and the macro level of society [69,98]. At the micro level (individuals), the SI initiatives analyzed aimed to improve people's well-being, by addressing the particular needs of specific vulnerable groups and related beneficiaries and actors (e.g., women farmers, fishermen, etc.) that are not met by the public or traditional private sectors (market). At the meso level (community), the initiatives analyzed improved the overall conditions of communities by, for example, promoting local development and strengthening community empowerment, as well as determining innovation in local governance and institutions. At the macro level (society), the SI initiatives analyzed have suggested new policy solutions, novel forms of social organization and ways of addressing problems. They have changed power relations between actors and sectors in society, promoting active political participation and citizen engagement, while contributing to the empowerment of citizens and communities. As noted in Section 6.2, they have not yet achieved systemic change.

## 7. Conclusions

This paper has assessed the impacts that nine SI initiatives related to the fields of agriculture, fishery, forestry, and rural development have had, or are having, on marginalized areas of Europe and the Mediterranean, by using a novel systematic evaluation framework [32]. The findings show the cross-sectoral (society, economy, environment, and government) nature and multi-level (micro level of individual, meso level of community, macro level of society) character of SIs. Nevertheless, SIs generated positive impacts, mainly inside the local territory in which they occur rather than outside it, showing no strong desire to go beyond the community. They also show that SIs contributed to address EU challenges and improved overall societal well-being, including reducing certain forms of marginality and supporting sustainability achievements.

These impacts can be considered to be providing a stabilizing effect that empowers local and regional stakeholders to further engage in SI initiatives and legitimate support of such marginalized areas by nation states and the European Union. Indeed, for SI to generate positive impacts that are sustainable and replicable, it is necessary to have a better support in terms of both policy and funds at national, regional, as well as at the EU level. We believe that in unlocking the potential of SI within marginalized communities, it is crucial to further assist bottom-up needs, to empower civil society and local actors (e.g., government bodies, NGOs) to act together within collaborative decision-making processes and innovative institutional/governance arrangements. It is also necessary to support more SIs with regional, national and EU funds and with financial instruments such as corporate social responsibility. Indeed, another way to finance SI is to receive funding from companies, which are now obliged to be agents of sustainable change in their communities, co-operating with local stakeholders in delivering social benefits. Also, for SIs to increase their critical mass, create stronger impacts, and become inspirational models, SI initiatives should be replicated in, or transferred to other areas. This requires the involvement of multiple actors across sectors at different administrative levels (e.g., global–local partnerships, cooperation among international organizations and private institutions and local stakeholders) and the inclusion of SI in main EU strategies and programs.

Practitioners and scientists across Europe provided evidence to the EU that SI should be boosted in marginalized territories because of the positive impacts and changes it can generate. This paper corroborates this message with empirical evidence.

**Author Contributions:** Conceptualization, E.R., C.D.T., E.P, L.S, R.D.R., V.M.G. and E.G.-M.; methodology, L.S., E.P., R.D.R. and V.M.G.; software, R.D.R. and V.M.G.; investigation, C.D.T., M.P., C.B, M.B., A.L. (Antonio Lopolito), A.L. (Arbia Labidi), A.B., A.V., S.S., M.N., M.D.-D. and N.P.; resources, C.D.T., M.P., C.B, M.B., A.L. (Antonio Lopolito), A.L. (Arbia Labidi), A.B., A.V., S.S., M.N., M.D.-D. and N.P.; formal quantitative analysis, R.D.R. and V.M.G.; data interpretation, E.R., C.D.T., M.P.,C.B, M.B., A.L. (Antonio Lopolito), A.L. (Arbia Labidi), A.B., A.V., S.S., M.M., M.D.-D. and N.P; data curation, C.D.T., R.D.R. and V.M.G.; writing–original draft preparation, E.R. and C.D.T.; writing–review and editing, M.P.,C.B, A.L., A.B., A.V., S.S., P.K., D.M., T.S., M.N., M.D.-D., N.P., L.S, E.P., E.G.-M., V.M.G. and R.D.R.; visualization, E.R., C.D.T., V.M.G., L.S. and E.P.; project administration and acquisition, D.M. and M.N. All authors have read and agreed to the published version of the manuscript.

**Funding:** This project has received funding from the European Union's Horizon 2020 Research and Innovation Program under Grant Agreement No. 677622 (H2020 SIMRA–Social Innovation in Marginalised Rural Areas Project).

**Institutional Review Board Statement:** The information regarding the nine SI initiatives were collected and analyzed within the framework of Horizon 2020 project SIMRA. Data collection and research design complied with the legal guidelines on research ethics as required under rules for receiving EU H2020 funding (Regulation (EU) 1291/2013, 11 December 2013), and the internal arrangements of individual project partners. The ethical clearance procedure is described in the SIMRA Deliverable 5.1 (http://www.simra-h2020.eu/wp-content/uploads/2018/06/SIMRA-D5.1 _Case-Study-Protocols-and-Final-Synthetic-Description-for-Each-Case-Study-1-1.pdf).

**Informed Consent Statement:** Informed consent was obtained from all subjects involved in the study.

**Data Availability Statement:** The data that support the findings of this study are available on request from the corresponding author (E.R.) starting from 2023. The data are not publicly available due to conditions associated with their collection (e.g., could compromise the privacy of research participants], in line with ethical clearance obtained. Further details of the case studies are available on the SIMRA project www site, www.simra-h2020.eu/index.php/simra-case-studies/.

**Acknowledgments:** This paper is based on research carried out as part of the Project Social Innovation in Marginalized Rural Areas (SIMRA) which is funded by the European Union's Horizon 2020 Research and Innovation Program under Grant Agreement No 677622. The views expressed in this article are the sole responsibility of the authors and do not necessarily reflect the views of the European Union. Input of the colleagues from the James Hutton Institute was also partly funded by the Rural and Environment Science and Analytical Services Division of the Scottish Government through its Strategic Research Programme (2016–2021).

**Conflicts of Interest:** The authors declare no conflict of interest.

## Appendix A

**Table A1.** Indicator values. (Note that unavailable values have been replaced by the median value).

| Category | Name | Meaning | WOM | SEA | VAZ | LUM | FIN | CAR | DAI | LAG | FLE | Median | Average | Min value | Max value |
|---|---|---|---|---|---|---|---|---|---|---|---|---|---|---|---|
| **A. Type of Impact** | A1 | Proportion of marginalization problems improved by the SI initiative | 1.00 | 0.00 | 0.00 | 0.00 | 1.00 | 1.00 | 0.43 | 0.59 | 0.75 | 0.59 | 0.53 | 0.00 | 1.00 |
| | A2 | Level of improvement in European societal challenges due to the SI initiative | 0.82 | 0.71 | 0.90 | 0.75 | 0.89 | 0.98 | 0.88 | 0.92 | 0.89 | 0.89 | 0.86 | 0.71 | 0.98 |

<div align="center">

**Table A1.** *Cont*.

</div>

| Category | Name | Meaning | WOM | SEA | VAZ | LUM | FIN | CAR | DAI | LAG | FLE | Median | Average | Min value | Max value |
|---|---|---|---|---|---|---|---|---|---|---|---|---|---|---|---|
| **B. Domains of Impacts** | B1 | Balance between positive to negative significant impacts due to the SI initiative in the four domains (social, economic, environmental, institutional) | 1.00 | 0.00 | 0.18 | 0.21 | 0.18 | 0.09 | 0.01 | 0.18 | 0.56 | 0.18 | 0.27 | 0.00 | 1.00 |
| | B2 | Proportion of the number of impacts of the SI initiative in the four domains which were positive | 1.00 | 0.86 | 1.00 | 1.00 | 0.83 | 0.67 | 0.97 | 0.97 | 1.00 | 0.97 | 0.92 | 0.67 | 1.00 |
| | B2_soc | Social | 1.00 | 1.00 | 1.00 | 1.00 | 0.90 | 1.00 | 1.00 | 1.00 | 1.00 | 1.00 | 0.99 | 0.90 | 1.00 |
| | B2_eco | Economic | 1.00 | 0.86 | 1.00 | 1.00 | 0.70 | 1.00 | 1.00 | 1.00 | 1.00 | 1.00 | 0.95 | 0.70 | 1.00 |
| | B2_env | Environmental | 1.00 | 0.75 | 1.00 | 1.00 | 1.00 | 0.00 | 0.00 | 1.00 | 0.00 | 1.00 | 0.64 | 0.00 | 1.00 |
| | B2_ins | Governance | 1.00 | 0.86 | 1.00 | 1.00 | 0.67 | 1.00 | 1.00 | 0.86 | 1.00 | 1.00 | 0.93 | 0.67 | 1.00 |
| | B3 | Level of effects of the SI initiative in the four domains inside and outside of the territory | 0.75 | 0.73 | 0.72 | 0.63 | 0.58 | 0.64 | 0.81 | 0.71 | 0.81 | 0.72 | 0.71 | 0.58 | 0.81 |
| | B3_soc | Social | 0.87 | 0.75 | 0.80 | 0.63 | 0.68 | 0.88 | 0.94 | 0.72 | 0.85 | 0.80 | 0.79 | 0.63 | 0.94 |
| | B3_eco | Economic | 0.78 | 0.83 | 0.73 | 0.63 | 0.50 | 0.69 | 0.96 | 0.70 | 0.75 | 0.73 | 0.73 | 0.50 | 0.96 |
| | B3_env | Environmental | 0.66 | 0.78 | 0.67 | 0.69 | 0.60 | 0.44 | 0.50 | 0.77 | 0.81 | 0.67 | 0.66 | 0.44 | 0.78 |
| | B3_ins | Governance | 0.68 | 0.55 | 0.58 | 0.58 | 0.56 | 0.56 | 0.81 | 0.58 | 0.86 | 0.58 | 0.64 | 0.55 | 0.81 |
| | B4 | Level of improvement in governance aspects due to the SI initiative. | 0.88 | 0.71 | 0.63 | 0.71 | 0.33 | 1.00 | 0.78 | 0.58 | 0.96 | 0.71 | 0.73 | 0.33 | 1.00 |
| **C. Scale of Impacts** | C1 | Level of effects of the SI initiative inside the territory in the four domains | 0.82 | 0.68 | 0.80 | 0.75 | 0.61 | 0.64 | 0.87 | 0.77 | 0.88 | 0.77 | 0.76 | 0.61 | 0.87 |
| | C2 | Level of effects of the SI initiative outside the territory in the four domains | 0.62 | 0.75 | 0.70 | 0.50 | 0.55 | 0.67 | 0.75 | 0.64 | 0.75 | 0.67 | 0.66 | 0.50 | 0.75 |

<div align="center">

**Table A2.** Indicator name with detailed explanation.

</div>

| Category of Impacts | | Name and Meaning of Indicator | Respondent Type |
|---|---|---|---|
| **A. Type of Impact** | A1 | Proportion of marginalization problems improved by the SI initiative, as perceived by stakeholders. The marginalization problems include: (i) physical geography constraints; (ii) infrastructural access limitations; (iii) socio-economic conditions. The reduction in the number of marginalization problems in the territory is measured by comparing the total number of problematic elements/aspects improved by the social innovation initiative with the problematic elements/aspects. | Stakeholders taking part in the focus group |
| | A2 | Level of improvement in European societal challenges due to the SI initiative, according to the actors. The indicators refer to the European Societal Challenges as identified in the Europe 2020 strategy (see https://ec.europa.eu/programmes/horizon2020/en/h2020-section/societal-challenges). The European societal challenges are: (i) health; (ii) ageing of population; (iii) income, jobs, education; (iv) sustainable agriculture and food security; (v) water use and quality; (vi) secure, clean and efficient energy; (vii) smart, green and integrated transport; (viii) environment and climate change; (ix) inclusive societies; innovative societies; (x) secure societies. | Core group; project partners |

**Table A2.** *Cont.*

| Category of Impacts | | Name and Meaning of Indicator | Respondent Type |
|---|---|---|---|
| **B.**<br>**Domains of Impacts** | B1 | Balance between positive to negative significant impacts due to the SI initiative in the four domains, according to perception of stakeholders.<br>The indicator measures the balance between the 4 greatest positive impacts and the 4 greatest negative impacts due to the Social Innovation initiative in the environmental, economic, social and institutional domains. Stakeholders attribute the scores based on the following four criteria: (i) capability of the social innovation to keep under direct control the impact; (ii) frequency of the activities determining the impacts; (iii) magnitude (intensity) of the impact; (iv) sensitivity of the local community to the impact. | Stakeholders taking part in the focus group |
| | B2 | Proportion of the number of impacts of the SI initiative in the four domains which were positive, according to perception of stakeholders.<br>The indicator measures the number of impacts of the social innovation initiative in the four domains. The elements refer to environmental, economic, social and institutional domains. For each domain, a detailed list of elements has been provided and analyzed by the stakeholders who participated in the focus group. | Stakeholders taking part in the focus group |
| | B2_soc | Social domain: life satisfaction and happiness, solidarity and mutual trust among the members of the community, civic engagement, safety and security of community members, food security, access to quality education for children and youths, options for life-long learning of adults, housing, welfare and social expenditure, gender balance, people at risk of poverty and social exclusion, vulnerable groups, health conditions and well-being of the members of the community, quality of service of general interest. | |
| | B2_eco | Economic domain: household income; investments on infrastructure that affects the community, investments on economic and social initiatives in the community; investments in research, experiments and innovation that increase knowledge; value added produced by the production, value chains; access to credit and insurance; wages of employees and workers; employment opportunities and quality; labor conditions in the sector in the territory. | |
| | B2_env | Environmental domain: pollutant emissions to air; carbon sequestration; water (e.g., consumption, quality); landscape and ecosystems; raw materials (e.g., wood, feedstock, fish); energy (e.g., consumption, percentage of renewable sources); biodiversity (e.g., animal and plant species, habitats, protected areas, genetic resources); soil (e.g., fertility, erosion, landslide stability); waste and/or effluents; noise or other types of disturbances (e.g., light pollution). | |
| | B2_ins | Institutional domain: stakeholders empowerment and representativeness in the decision-making process; capability of public administrations to manage collaboration, dialogue and/or conflicts; capability of the community and public administrations to adapt to crises and disturbances; coherence of local policies and actions with international and national policies and actions; legality; transparency and open access to data and knowledge sharing; accountability of both private and public organizations; trust in public institutions; professional capability of public officials and administrations. | |
| | B3 | Level of effects of the social innovation initiative in the four domains, according to the actors.<br>The indicator measures the extent of the effects of the social innovation initiative inside and outside the territory in the four domains. The indicator is based on a Likert Scale from -2 (negative) to + 2 (positive) in relation to four domains (economy, social cohesion*, public administrations* and the environment). | Core group; project partners; beneficiaries. |
| | B3_soc | Same as B2_soc | |
| | B3_eco | Same as B2_eco | |
| | B3_env | Same as B2_env | |
| | B3_ins | Same as B2_ins | |
| | B4 | Level of improvement in governance aspects due to the SI initiative, according to the actors.<br>The indicator measures the level of improvement in different aspects of governance due to the social innovation initiative, as perceived by the innovator(s), follower(s) and project partners. Respondents score the improvement for 13 aspects of governance: (i) options for citizen engagement; (ii) stakeholder consultation; (iii) voice of minorities; (iv) gender balance; (v) transparency; (vi) bureaucracy; (vii) capacity of public administrations; (viii) policy initiatives; (ix) legal framework; (x) conflict of interests and corruption; (xi) quality of public services; (xii) market and economy; (xiii) other. | Core group; project partners |
| **C. Scale of Impacts** | C1 | Level of effects of the SI initiative inside the territory in the four domains, according to the actors.<br>The indicator measures the extent of the effects of the social innovation initiative inside the territory in the four domains. The indicator is based on a Likert Scale from -2 (negative) to + 2 (positive) in relation to: (i) economy; (ii) social cohesion; (iii) public administrations; (iv) the environment. | Core group; project partners; beneficiaries. |
| | C2 | Level of effects of the SI initiative outside the territory in the four domains, according to the actors.<br>The indicator measures the extent of the effects of the social innovation initiative outside the territory in the four domains. The indicator is based on a Likert Scale from -2 (negative) to + 2 (positive) in relation to: (i) economy; (ii) social cohesion; (iii) public administrations; (iv) the environment. | Core group; project partners; beneficiaries. |

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
