# Peer review of "Can Social Innovation Make a Change in European and Mediterranean Marginalized Areas? Social Innovation Impact Assessment in Agriculture, Fisheries, Forestry, and Rural Development"

_sustainability, doi:10.3390/su13041823_

Round 1

Reviewer 1 Report

The submitted manuscript concerns the current and important topics related to Social Innovation (SI) impact assessment in agriculture, fisheries, forestry, and rural development.

The structure of the paper is correct. The article is divided into logically following chapters and subchapters.

The purpose of the article is clearly articulated in the Introduction chapter.

Theoretical Background sufficiently introduces the reader to the discussed issues. The literature is well chosen.

The research methods are correctly selected and the research process is described in an understandable way.

I have only two remarks that I propose the Authors consider whether they are worth addressing. I list them below:

1) Linie 652-654: "These empirical data confirm what is described in the existing literature as promoting social innovation in rural areas because of its positive effects on economy and society [eg, 71,10,48,72,73,12,74, 75.76.65.77] "
I suggest describing this issue in more detail, and not just such a laconic statement.

2) The Conclusions chapter is fairly general. I would suggest extending it.

To sum up, the article is written in an understandable language, it is easy to read, it concerns the current and important subject matter both in terms of science and practice.

As for linguistic correctness, I do not feel competent to express myself because English is not my mother tongue.

Author Response

1) Linie 652-654: "These empirical data confirm what is described in the existing literature as promoting social innovation in rural areas because of its positive effects on economy and society [eg, 71,10,48,72,73,12,74, 75.76.65.77] "I suggest describing this issue in more detail, and not just such a laconic statement.

Response 1: I thank the reviewer for this comment. I have better explained in which sense SIs provide positive benefits for the communities in which occur mainly considering the social, economic and institutional domains.

2) The Conclusions chapter is fairly general. I would suggest extending it.

Response 2: I thank the reviewer for this comment. I have extended the conclusions by better explaining how SI should be better supported and replicated.

Reviewer 2 Report

Dear Authors, 

Please find attached my Comments and Suggestions for you.

Good luck with your work!

Kind regards,

The Reviewer

Author Response

I thank very much the reviewer for the constructive comment. I have introduced into the paper references to how Social Innovation is addressing sustainability challenges and promoting sustainable achievements for the communities in which occurs (see introduction, discussion and conclusion). Also, I made references to CSR as a tool companies use to achieve sustainability (see introduction) and that can also be used to finance SI (see conclusion).

Round 2

Reviewer 2 Report

Dear Authors,

Thank you for the revised version of your Article!

Also, thank you kindly for your Comments and Notes!

I wish you good luck with your work!

Kind regards,

The Reviewer